# Cyrcadian Rhythm, Mood, and Temporal Patterns of Eating Chocolate: A Scoping Review of Physiology, Findings, and Future Directions

**DOI:** 10.3390/nu14153113

**Published:** 2022-07-28

**Authors:** Sergio Garbarino, Emanuela Garbarino, Paola Lanteri

**Affiliations:** 1Department of Neuroscience, Rehabilitation, Ophthalmology, Genetics and Maternal/Child Sciences (DINOGMI), University of Genoa, 16132 Genoa, Italy; sgarbarino.neuro@gmail.com; 2Department of Psychology, University Milan-Bicocca, 20126 Milan, Italy; e.garbarino@campus.unimib.it; 3Neurophysiopathology Center, Fondazione IRCCS Istituto Neurologico Carlo Besta, 20133 Milan, Italy

**Keywords:** chrononutrition, circadian rhythms, chocolate, flavonoid, brain function, mood

## Abstract

This paper discusses the effect of chrononutrition on the regulation of circadian rhythms; in particular, that of chocolate on the resynchronization of the human internal biological central and peripheral clocks with the main external synchronizers, light–dark cycle and nutrition-fasting cycle. The desynchronization of internal clocks with external synchronizers, which is so frequent in our modern society due to the tight rhythms imposed by work, social life, and technology, has a negative impact on our psycho-physical performance, well-being, and health. Taking small amounts of chocolate, in the morning at breakfast at the onset of the active phase, helps speed up resynchronization time. The high flavonoid contents in chocolate promote cardioprotection, metabolic regulation, neuroprotection, and neuromodulation with direct actions on brain function, neurogenesis, angiogenesis, and mood. Although the mechanisms of action of chocolate compounds on brain function and mood as well as on the regulation of circadian rhythms have yet to be fully understood, data from the literature currently available seem to agree in suggesting that chocolate intake, in compliance with chrononutrition, could be a strategy to reduce the negative effects of desynchronization. This strategy appears to be easily implemented in different age groups to improve work ability and daily life.

## 1. Introduction: Life, Health, and Circadian Rhythms

The Earth’s rotation around its axis causes periodic light and dark variations in the environment over a 24-h period. These predictable variations in the light environment allow organisms to optimally organize their physiology and activity-rest rhythms to specific periods of the day-night cycle [1]. Indeed, there is a clear selective advantage in anticipating changes in the environment, particularly light–dark, by adapting their activities. Therefore, evolved organisms have an internal biological clock that functions with a period close to 24 h, which is circadian, in the absence of environmental influences such as light [2]. Anticipation of daily events organized in this way allows optimal management of time and available energy, giving those organisms a significant advantage [3]. The correct alignment between the light–dark rhythm, the circadian clock, and behavior determines a temporal order in organisms that is of fundamental importance for ensuring survival [4].

The circadian clock drives many outputs, including sleep–wakefulness, hormonal, and metabolic rhythm. Locomotor activity and sleep are segregated into specific phases of the light–dark cycle [5]. Endocrine processes are also regulated by a rhythmic periodicity: the cortisol concentration in circulation reaches a peak level in the morning, and unlike many hormones involved in metabolism, such as leptin and ghrelin, reach their peak at night [6]. Urine production has a circadian nature [7], as well as perception–nociception [8], temperature [9], vigilance [10], and cognitive abilities such as memory and mathematical performance [11,12]. Mood also appears to have a circadian cycle [13,14], with more evidence supporting a role in positive affect [15].

The light–dark cycles drive the central clock located in the hypothalamic suprachiasmatic nucleus (SCN), where it mainly dominates the rhythms related to activity–rest oscillations. SCN favors the active phase by acting on the sympathetic nervous system, anticipating the rise in body temperature and blood pressure. The same model also applies to the feeding–fasting systems that prepare/activate before the activation phase, breakfast, according to the time of local clocks and guide the peripheral clocks present in most tissues and, to some extent, in the brain [16]. Indeed, peripheral clocks dominate local physiological processes, such as glucose and lipid homeostasis, hormone secretion, xenobiotics, immune response, and the digestive system [16]. Since the central clock organizes local clocks through neuronal and humoral signals, the mismatch between the central and peripheral clocks results in a condition of desynchronization [17]. The desynchronization condition contributes to the sleep disorders, adverse metabolic effects of circadian misalignment, such as increased risk of type 2 diabetes among shift workers [18], cancer, and psychological/psychiatric disorders [19].

In this paper, we aim to review the available data in the literature on the role of nutrition as a synchronizer of biological clocks with regard to chocolate and its mechanisms of action on brain function and mood.

## 2. Nutrition: Not Only Light Acts as (De)Synchronizer

While it has long been recognized that the light–dark cycle is the main zeitgeber for the central clock in the brain, providing input to central clock genes (CLOCK, BMAL, PER1, PER2, PER3, CRY1, CRY2, Tim) in central nervous system cell populations, food intake may act as an important time synchronizer for tissue metabolic clocks, but not for the brain [20]. Temporal signals from the SCN drive circadian rhythms of feeding–fasting and activity–rest, which in turn act by synchronizing clock genes in peripheral clocks [21], such as liver cells and skeletal muscle, the heart, white adipose tissue, and other metabolic tissues.

The timing of food intake is a key factor in determining the phase state of peripheral circadian clocks. As such, it is a powerful signal of activation of peripheral oscillators involved in metabolic and digestive functions, such as liver, kidney, intestine, and adipose tissue [22]. 

Moreover, although food exerts a strong synchronizing effect on peripheral clocks, it also exerts a function at the brain level by acting mainly on brain areas involved in energy balance and motivation/reward for food intake [23,24,25]. It is important to note that the two main synchronizers, feeding–fasting and light–dark, act differently on circadian rhythms in different systems and organ tissues, as the signals provided by the light–dark cycle act predominantly on central systems and certain brain regions, including the SCN, while feeding–fasting acts mainly on peripheral systems [26,27]. In fact, restricted food intake at certain times during the day for one week is able to completely alter the expression phase of genes controlled by the circadian clock in the peripheral tissues of nocturnal rodents, whereas it does not affect the central clock, which is dominated by light–dark cycles [27,28,29].

Among peripheral clocks, the liver adapts most rapidly to changes in feeding–fasting rhythms, within about 3 days, whereas the kidney, heart, pancreas, and lung take longer [28,29].

Among different feeding times, studies in mice mimicking human dietary patterns have shown that breakfast is usually the most effective meal in determining liver clock phase because breakfast is taken after the longest fasting phase of the day [30]. Therefore, late meals or midnight snacks distort the hunger period and significantly modify the phase of peripheral clocks [22].

## 3. Chrononutrition and Food Components Affecting Circadian Clocks

Both central and peripheral circadian clocks can be influenced by the consumption of different food components by different mechanisms. 

The reciprocal influences of circadian clocks and energy metabolism suggest that feeding time has a critical impact on metabolism. The quality and quantity as well as the timing of food intake are all important for nutrition: this is referred to as ‘chrononutrition’. Indeed, food intake has a knock-on effect on peripheral clocks, mainly the liver, but not on the SCN, so that food intake outside the physiologically designated ‘time’ window induces desynchronization between peripheral clocks and SCN [29]. 

Tissue-specific gene expression of peripheral clocks is regulated by food intake, whereas the central gene expression of the SCN itself is protected by nutrient deregulation [20]. Timing of food intake also influences body weight and obesity risk in humans [31,32]: a two-group study evaluating weight loss during an isocaloric diet showed improvements in several metabolic indicators such as body weight, fasting blood glucose, insulin, ghrelin, average hunger, and satiety scores in the group eating a larger breakfast and lighter dinner compared to the other way around [33]. In both humans and experimental animals, eating dinner late at night and skipping breakfast leads to increased body weight and obesity [3].

With regard to individual food components associated differently with different eating times, it has been shown that eating a diet rich in fat *ad libitum* attenuates the amplitude of the clocks, while eating a diet limited in time restores their amplitude by regulating the circadian cycle. A combination of carbohydrates and protein also appears to be essential for proper resetting of peripheral clocks such as the liver clock [34]. 

In contrast, the intake of protein nutrients at times of phase-shifting hepatic cell clocks correlates with blood glucose absorption, and readily digestible starches with a high glycemic index have a powerful entrainment effect on peripheral liver clocks [35]. 

The action of specific nutrients occurs through the different expression of clock-controlled genes [36]. Glucose is able to regulate BMAL1 and period expression [37] as demonstrated, for example, in a study of mice fed a high-fat diet in which an alteration of lipid metabolism genes was documented [38]. Indeed, this finding is further corroborated by a study conducted by Eckel-Mahn et al., which illustrated that most hepatic metabolites also follow changes according to a circadian rhythm, and that these variations are regulated by both the clock transcriptome and the feeding–fasting cycles that favor the maintenance of hepatic homeostasis [39].

A high-fat diet, such as the ketogenic diet with a high fat content and low carbohydrate content, has the greatest effect on the central clock and/or eating behavior: a high fat content reduces the duration of the circadian rhythm of locomotor activity under conditions of constant darkness, whereas under normal conditions of alternating light–dark, there is an advancement in the rhythm of clock expression and clock-controlled genes in peripheral tissues [40]. In recent years, the field of nutraceuticals and functional foods has developed alongside interest in the potential modulatory effects of food constituents on human health. In this context, mention should be made of caffeine/theophylline, which could influence our circadian system, and flavonoids. Caffeine in food and drink has been shown to prolong circadian locomotor rhythms in Drosophila and mice [41,42]. The dose of caffeine needed to influence circadian rhythms appears to be low, at 0.05%, equivalent to the dose contained in coffee. Indeed, consumption of non-decaffeinated coffee prolongs the rhythm of circadian activity in mice under constant dark conditions [42]. Interestingly, both caffeine and theophylline also prolong the circadian period in cultured cells, mouse tissues, Neurospora, Chlamydomonas, Drosophila, and even higher plants [37,43,44,45]. 

Among the dietary constituents, flavonoids, a class of polyphenolic compounds, have received particular interest for their various beneficial biological actions, such as neurological and cardiological protection and neuromodulation [46]. In particular, a rich source of flavonoids has been detected in the cocoa bean, especially the subclass of flavanols in the form of epicatechin and catechin [47].

### 3.1. Chocolate

Cocoa products and especially chocolate are foodstuffs originating from South America. Cocoa is obtained from the seeds of the Theobroma cacao tree, which are then dried, shelled, fermented, and ground with other substances such as sugar, fat, and other flavorings to produce the wide variety of chocolate available on the market, from dark to milk variations.

Chocolate consists of a combination of several ingredients, the main ones being cocoa, cocoa butter, and sugar, which make up the solid food product. Cocoa beans, along with foods such as tea, red wine, and fruit, are a rich source of flavanols, a subgroup of natural flavonoids that are bioactive plant compounds.

There have been reports of the health benefits of chocolate since ancient times, the earliest dating back to Aztec and Mayan medical practice [48]. It was not until the end of the 20th century, however, that claims about the alleged health benefits of chocolate attracted more and more scientific interest. Flavonoids, contained in cocoa, have been approved by the European Food Safety Agency as being beneficial to health [49]. 

Due to dark chocolate’s high concentrations of flavanols, a subgroup of flavonoids, especially epicatechin, [50] known as powerful antioxidant agents [51], have received a health claim in relation to their impact on ‘maintaining normal endothelium-dependent vasodilation’.

Many studies conducted so far focus on the effects of chocolate on the cardiovascular system, skin, cholesterol levels, the release of the neurotransmitters anandamide and serotonin, and on the properties of specific stimulating constituents such as theobromine and caffeine [52].

#### 3.1.1. Biochemical Components and Neurobiological Impact

Chocolate contains more than 300–500 known chemicals, some of which also act on brain cells and modulate mood [53]. Of these 300 to 500 chemicals in chocolate, some play an important role in humans, influencing neurocognitive functions. 

The main psychoactive components of chocolate are [54,55] as follows. 

-Carbohydrates, which have known behavioral effects.-Flavanols, which are ubiquitous in the plant kingdom. In foods normally consumed in the diet, high levels of flavonoids can be found in green and black tea, grapes, red wine, apples, and especially in cocoa and cocoa-containing products. In fact, cocoa is particularly rich in flavonoids and contains a distinct complement of flavanols (a subclass of flavonoids), flavan-3-ols, mainly present in the form of epicatechin and catechin [50], and their derivatives in high concentrations [56]. Flavan-3-ols are the building blocks for polymeric procyanidin type B-2.-Methylxanthines (MX), such as caffeine and its highly fat-soluble derivative and metabolite theobromine, which have peak plasma levels 60–120 min after ingestion.Like caffeine, theobromine binds to adenosine receptors, exhibiting its psychoactive potential similar to that of caffeine. However, these two MX have distinct functional binding properties.-Biogenic amines, such as serotonin, tryptophan, phenylethylamine, tyrosine, tryptamine, and tyramine, have a concentration that increases during fermentation and decreases during roasting and alkalinization.-Anandamide, an endogenous ligand for the cannabinoid receptor that is found in low quantities, such as 0.5 mg g^−1^, salsolinol, and tetrahydro-b-carboline.

The most studied substances are flavanols and their metabolites: it has been shown in animal studies that these products can pass through the blood-brain barrier (BBB), with positive effects on brain tissue, vessels, and function (angio- and neurogenesis, changes in neuron morphology), stimulating cerebral blood circulation [57]. Epicatechin [50], the most common flavanol found in cocoa, is rapidly absorbed by humans. Thirty min after intake it is detectable in blood plasma, and its peak is reached after 2–3 h, showing a concentration directly related to the dose of chocolate ingested [58], and then it returns to basal level within 6–8 h after consumption.

#### 3.1.2. Chocolate and Brain Functions

In order to exert effects in the brain, flavanols cross the BBB by a process that is not only time-dependent but also stereoselective, which, thus, favors the passage of epicatechin more than catechin, as has been demonstrated in two cell lines, one of rat and one of human origin [59]; the permeability is proportional to the degree of lipophilicity and inversely proportional to the degree of polarity.

In humans, the main polyphenols increase cerebral blood flow (CBF) and are derived from nutrients such as cocoa, wine, grape seeds, berries, tea, tomatoes, and soy [60].

Optimal brain function requires an adequate CBF that promotes the correct and constant supply of oxygen and glucose to neurons and the excretion of deposited waste products. Increasing CBF is also a potential tool for enhancing brain function. Flavanols and their metabolites have the ability to reach and accumulate in the brain regions mainly involved in learning processes and memory, and are therefore thought to exert a direct brain action on cognitive function and neuroprotection [55]. After chronic administration of chocolate, high concentrations of tangeretin were found in the striatum nucleus, hypothalamus, and hippocampus of the rat [61].

The neurobiological impact of flavanols on the brain in the areas of learning, memory, cognitive function, and mood is thought to occur mainly in two ways. The effects of flavonoids on the brain are mediated by their function of neuroprotecting vulnerable neurons in particular, improving neuronal function and stimulating regeneration (neurogenesis) [62] by interacting with intracellular neuronal signaling pathways that control neuronal survival and differentiation, long-term potentiation (LTP), and memory.

First, flavonoids interact with a number of cellular signaling pathways by activating gene expression and protein synthesis for the maintenance of LTP, and the stabilization of long-term memories [63] are critical for neurogenesis, synaptic growth, and survival of neurons, mainly in the brain hippocampus and subventricular area related to learning and memory [64,65].

Secondly, flavonoids induce vasodilation through nitric oxide at both cardiovascular and peripheral levels through the production of nitric oxide (NO), a key regulator of vascular function, which acts as a signaling molecule by inhibiting the action of adhesion molecules in atheromatous plaque that cause inflammation [66], and, most importantly, promote and improve the function of the vascular endothelium by acting, with dilating action, on the smooth muscle tissue of blood vessels [67]. This effect on the vascular system with endothelium-dependent vasodilation, contributes to the maintenance of normal blood flow and improvement of blood pressure; it also induces a reduction in platelet aggregation [68,69,70,71,72,73]. This. in turn. results in increased CBF and blood perfusion throughout the central and peripheral nervous system [74], allowing better oxygen and glucose delivery to neurons and removal of waste metabolites in the nervous systems [75]. Among flavanols, epicatechin has the greatest ability to increase nitric oxide (NO) bioavailability, leading to improvements in vascular tone and blood pressure regulation [76]. These vascular changes occurring at the peripheral level may also extend to cerebral perfusion, leading to optimized cerebrovascular integration during neuronal activation phases, a mechanism considered crucial for the functional and structural integrity of the brain and for promoting adult neurogenesis in the hippocampus [77].

Administration of cocoa flavonoids, therefore, also stimulates angiogenesis in the hippocampus [51,78], as demonstrated by administering epicatechin, given to mice at a dose of 500 mg g^−1^ (daily supply of 2.5 mg). Combining epicatechin with exercise also improved the consolidation of spatial memory and the density of dendritic spines in the dentate gyrus of the hippocampus. In the same study, epicatechin treatment was shown to increase learning-associated gene expression in the hippocampus, while it did not appear to influence neurogenesis in the adult hippocampus [78].

Thus, flavonoids could also exert their neurocognitive effects both directly and indirectly by interacting with the cellular network and molecular system deputed to memory acquisition, storage, retrieval, and learning [62], also through long-term potentiation, synaptic plasticity [79], enhanced neuronal connection, and communication.

Epidemiological studies suggest that regular flavonoid intake may be associated with better cognitive function [80], to decreased risk of dementia and cognitive decline [81,82,83], better cognitive development over a 10-year period [84], and improved dose-dependent cognitive performance in physiological aging [85].

#### 3.1.3. Chocolate and Mood

A lot of data from the literature support the hypothesis of the influence of theobromine and caffeine on mood and cognitive function [86,87,88,89], but the impact and mechanism by which flavanols affect mood remains unclear.

It is commonly believed that eating chocolate improves mood and rapidly induces a sense of well-being in people [89]. An initial rapid effect of chocolate on emotional comfort appears to be related to the ability of the carbohydrates it contains, to promote such positive feelings through the release of several gut and brain peptides [90].

In rats, the intake of cocoa-extracted polyphenols, while significantly reducing the duration of immobility in a forced swimming test, had no effect on locomotor activity in the open field, confirming its specific antidepressant effect [91]. The most likely basis for this effect may be attributable to endorphin release [92]. Indeed, sweet food intake is increased by opioid agonists and decreased by opioid antagonists [93,94]. The effect of chocolate is also exerted through interaction with neurotransmitters such as dopamine (tyrosine contained in chocolate is the precursor to dopamine), serotonin, and endorphins, which contribute to appetite, reward, and mood regulation. The dopaminergic system contributes to the desire to consume chocolate, probably by acting mainly non-specifically towards food. After carbohydrate ingestion, only when the protein component of the meal is less than 2% does it induce an increase in serotonin concentrations in the brain [90]. It should be noted that chocolate contains 5% of its caloric content in the form of protein, which would cancel out any effect of serotonin. Moreover, manipulations of tryptophan, the precursor of serotonin, also cause physiological changes that are too slow to explain the mood effects described during or immediately after eating chocolate [95]. Another area where chocolate might act could be in the area of opioids, which are known to play a role in the palatability of preferred foods [96], releasing endorphins during food intake, and, thus, justifying the increase in pleasure during food intake [97]. The mood effects of cocoa may also be partly due to opioids released in response to the ingestion of sweets and other pleasantly palatable foods [98,99]. The increase in central opioidergic activity, in turn, stimulates the immediate release of beta-endorphin in the hypothalamus, which exerts an analgesic effect. Bad mood stimulates consumption of comfort foods such as chocolate in two different ways [100]. The former is called craving and is associated with an impulsive desire for chocolate, and its compulsive consumption occurs especially when under high emotional stress, showing a clear link between the perception of a negative mood and the intense desire to consume chocolate [101]. The association between chocolate craving and consumption under emotional stress was demonstrated in a study in which subjects had to listen to music that induced a happy or sad mood. Chocolate consumption was increased by listening to sad music [100].

The second modality to be considered is the palatability of the food. The pleasure induced by palatable food is regulated by endogenous opioids that stimulate food intake in rats. The pleasure induced by palatable foods is regulated by endogenous opioids that stimulate their intake in rats. In humans, however, the critical factor in satisfying chocolate craving appears to be taste and mouthfeel [102]. Females, mainly in the perimenstrual period, seems more sensitive to chocolate. The response to satiety seems to vary by gender [103].

It is more conceivable that an important role in liking or craving chocolate is due more to the composite sensory properties of chocolate than to its role in appetite and satiety [104]. During the consumption of chocolate, different brain areas are also activated depending on the motivation to eat chocolate, based on positive/appetitive stimuli or associated with negative/adverse stimuli. Modulation of brain activity has been observed in chemosensory cortical areas such as the insula, prefrontal regions, and caudomedial and caudolateral orbitofrontal cortex, with overlapping and co-activation under contrasting motivational conditions [105]. The ability to activate images of appetizing foods involved in food motivation and hedonism in a fronto–striatal–amygdala–midbrain network appears to be dependent on individual variability in reward sensitivity. If this same neuronal circuit is stimulated in the animal, it may result in the cancellation of the sense of satiety and cause overeating of highly palatable foods [106].

The smell of chocolate itself is sufficient to modulate brain activity recorded on the electroencephalography (EEG). The smell of chocolate induces a significant reduction in theta activity compared to any other stimulus. Theta activity is considered to be closely related to attentional level, cognitive load in general, and, in this specific case, to olfactory perception, so a reduction in theta activity could be indicative of a reduced level of attention and an increased propensity to distraction [107]. In addition to olfaction, the sight of chocolate also evokes activations in the brain and especially in the medial orbitofrontal cortex and ventral striatum, particularly in subjects who crave chocolate compared to those who do not: the combination of an image of chocolate with chocolate in the mouth evoked greater brain activation than the sight of the sum of the different components in the medial orbitofrontal cortex and cingulate cortex [108].

The motivation for chocolate preference appears to be primarily, if not entirely, sensory. The origin of the liking of its sensory properties is unclear; it could be innate or acquired based on the sweetness, texture, and aroma characteristics of chocolate, or it could depend on the interaction between a person’s state and the post-gastronomic effects of chocolate. Surprisingly, there is little evidence of a relationship between chocolate addiction and chocolate liking [102]. However, chocolate consumption fails to activate the key structure for drug addiction, the nucleus accumbens [109,110,111].

The effect of chocolate on mood may be attributed to the affinity for adenosine and benzodiazepine (GABAa) receptors of polyphenolic compounds, which means that their ingestion may have a soothing effect [112]. Some polyphenolic compounds indeed have anxiolytic properties [113]. A small randomized controlled pilot study in humans with chronic fatigue reported a reduction in anxiety-related symptoms after eating polyphenol-rich chocolate, compared to polyphenol-poor chocolate [114].

Pase et al. [115] investigated the acute and subchronic effects of polyphenol supplementation on mood and cognitive performance in a randomized, placebo-controlled, double-blind study. Thirty days of treatment with a high dose of cocoa polyphenols reduced self-rated anxiety and contentment. No significant effect on cognitive performance was recorded with either the high or low dose in either the acute or chronic phase.

The optimal dosage of cocoa polyphenols needed to improve cognitive function and mood remains unclear.

#### 3.1.4. Chocolate, Sleep and Circadian Rhythms

Our modern society functions at a hectic pace of activity 24/7, which leads individuals to sacrifice sleep hours and disregard daily sleep–wake rhythms. As experienced daily by shift workers, jet-lagged travelers, or those with so-called social jet-lag syndrome, disturbed sleep–wake rhythms create a conflict, a temporal mismatch between the circadian system and temporal signals derived from the cyclical environmental changes, such as the light–dark cycle, [116] or a desynchronization [117]. This condition, if prolonged over time, leads to chronic sleep disorders (CSD) that result in deficits to health and psychophysical well-being. The most frequent symptoms belong to the neurobehavioral sphere and are often associated with mood changes such as a tendency to depression and impairment of cognitive functions, especially executive functions, as well as cardio- and cerebrovascular disorders, stroke, hypertension, obesity, and diabetes [118]. CSD are a socio-economic and public health issue due to their high prevalence in the general and working population, their impact on health, and work output given the higher incidence of absenteeism, and increased rates of errors and accidents at work [119]. In addition, CSD often have a bidirectional relationship with stress [120].

Jet lag results from a sudden change in the light–dark cycle due to trans-south travel or social life (in the case of social jet lag), which leads to a misalignment between internal circadian rhythms, mainly but not limited to SCN, and the external rhythmic time-cues (Zeitgeber), mainly light, for the day–night cycle. The days required for the process of coordinating the internal circadian clock to external rhythmic are often associated with behavioral and physiological discomfort.

Escobar C. et al. [121] observed that chocolate administration resulted in a faster rate of realignment and synchronization between activity–rest cycle and circadian temperature rhythms. Programmed access to chocolate activates brain areas involved in motivation and metabolic response to food [23] as well as the circadian system by improving neuronal activation in the SCN [122]. Other studies have also reported a similar effect on the speed of realignment to feeding schedules in peripheral oscillatory clocks [26,27,123].

However, it is not only the type of food that determines the ability to realign and re-synchronize internal clocks with external signals but also the time of intake of food, especially for palatable food.

Due to this complex interaction between external Zeitgeber and internal circadian rhythms, the greatest beneficial effects of entrainment on circadian function are seen when food intake coincides with the activity phase [124,125], whereas an inhibitory effect occurs when food is taken during the rest phase [126,127] (Figure 1).

Furthermore, to maintain a coordinated and synchronized circadian function, food intake must be phased with the light–dark cycle. The main effect seems to be due to a direct synchronization action on brain oscillators and central and peripheral clocks [26,128].

Under normal light and dark (LD) conditions, programmed food intake does not shift the SCN phase [27,129]. Other studies indicate that SCN is inhibited during food anticipation and fasting as observed with c-Fos, a major early gene that is activated by external signals [130,131,132] or electrophysiological recordings, whereas the ventral SCN is activated both after re-feeding and with light [133].

Recent results indicate that the SCN may also respond to palatable food construed as hedonic information, via dopaminergic projections from the ventral tegmental area [134]. The rapid achievement of synchronization with limited daily chocolate intake may also be partly due to the increase in arousal induced by chocolate intake as a hedonic effect.

When planned for breakfast, an appetizing food, such as chocolate, can influence activation in the SCN, at the level of the dorsomedial region [121]. This rhythmic pattern in the dorsal SCN may promote faster re-entrainment [135] when bounded by a time window in which chocolate was administered during the active phase, whereas chocolate did not promote re-entrainment when administered during the resting phase.

Scheduled feeding has been shown to be a strong entrainment signal for circadian rhythm; especially when food intake is in phase with the period of activity. This exerts beneficial effects on the circadian system by favoring its synchronization and activation of the metabolism [136,137], as demonstrated in experimental studies on shift-worker models. Time-limited access to food accelerates resynchronization in a jet-lag model, prevents circadian desynchronization in a shift-work model, and induces positive effects in metabolism [138,139]. In contrast, food scheduled in the sleep–rest phase slows circadian synchronization and metabolism and alters behavior [138,139,140].

Recently, Oishi et al. [141] confirmed the positive action of cocoa on sleep disturbance induced by psychophysiological stress in mice using EEG. Cocoa intake attenuated the alteration of circadian sleep–wake rhythms. The EEG revealed that cocoa significantly improved both the increase in the level of alertness during the first half of the light period and the increase in NREM sleep during the first half of the dark period in mice with CSD. Under non-CSD conditions, cocoa does not appear to influence either the rhythms of run-rest activity or sleep–wake cycles. It is hypothesized that this positive action may be attributable to high concentrations of flavonoids in cocoa (epicatechin, catechin, and procyanidins), which improve blood flow and have antioxidant and neuroprotective properties. Indeed, in experimental animals, sleep deprivation and CSDs in general have been shown to increase oxidative stress levels in specific brain regions such as the hypothalamus, hippocampus, and thalamus [142].

Flavanols acting on endothelial function could also play a role in insomnia, as it is just endothelial dysfunction that appears to be responsible for some insomnia-related symptoms and the association between insomnia and cardiovascular disease [143]. Cocoa flavanols, by facilitating nitric oxide production, improve vascular endothelial function due to their vasodilatory effect [51]. CBF also appears to play an important role in sleep regulation [144] as well as cognitive and emotional processes, although it is not known how cerebral blood flow varies during alternating sleep–wake cycles [145]. Flavanol-rich cocoa significantly increases cerebral blood flow in humans [46,51] and attenuates the CSD-induced disturbance of circadian activity rhythm, sleep–wake cycles, cognitive functions by improving cerebral endothelial cell function, and blood flow, as demonstrated by Grassi et al. [76].

In addition, another pathway through which cocoa has a protective effect on the synchronous maintenance of the sleep–wake rhythm in subjects with CSD is the modulated neurotransmission of serotonin [146,147]. In fact, regular cocoa consumption has been shown to increase serotonin concentrations in the brain [148].

Acute administration of flavanol-rich chocolate can counteract the negative effects of total sleep deprivation both on working memory performance in healthy young people [76] and on endothelial and arterial function and, thus, on blood pressure (Figure 1). Natural cocoa seems to be an ideal nutrient for ameliorating stress-induced psychophysiological sleep disturbance without distorting behavioral or sleep regulation under normal conditions.

## 4. Conclusions

Modern society imposes increasingly stressful rhythms of life and work, and technological innovations have led us to live an active life 24/7. The immediate effect is on the quality and quantity of sleep, which in turn has repercussions on our lower tolerance to stress, alterations in mood, and greater susceptibility to infectious, metabolic, and cardiovascular diseases. Lifestyles are undoubtedly the main targets to work on to maintain a harmonious synchronization between our central and peripheral endogenous clocks and external synchronizers. Of course, the circadian rhythm of light and dark is the most powerful synchronizer, but we must not forget the other fundamental synchronizer: food and fasting.

Chocolate is a food that, due to its high flavonoid content, when taken in the morning, can have positive effects on our mental and cognitive well-being, our cardiovascular system, and our metabolism. Chocolate also induces positive effects on mood and is, therefore, often spontaneously consumed under conditions of emotional stress. In addition to the beneficial effects on the vascular system and cerebral blood flow, flavonoids have a protective function on the neuronal cell by inhibiting neuronal death by apoptosis induced by neurotoxicants, such as oxygen radicals through interaction with signaling cascades involving proteins and lipid kinases. They also promote neuronal survival, neurogenesis, and synaptic plasticity [89], preserving cognitive abilities during the stages of aging. Crossing the BBB, they act on both the vascular and neurocellular sides of the brain, on one hand, by stimulating cerebral perfusion and promoting angiogenesis, and, on the other, by modulating changes in the morphology of the neurons participating in learning and memory processes. All these properties are of great interest, but it is currently unclear when consumption of flavonoid-rich cocoa and chocolate should be initiated to achieve beneficial and protective effects against the mechanisms underlying cognitive decline and age-dependent neurodegenerative diseases [52,149]. Many studies are still needed to understand the mechanisms of action and the timing of the neuroprotective activity of cocoa and chocolate.

Although the dosages to be taken in order to achieve the beneficial effects are still unclear, the literature data support the suggestion that the beneficial effect is obtained by eating cocoa only in modest amounts, preferably dark chocolate, during the activity hours of the day and in the early part of the day to avoid problems of weight gain. Frequent consumption of chocolate managed in this way might actually be associated with a lower body mass index [150].

Time of food intake is now suggested as a chronotherapeutic strategy that may help speed up the time needed to resynchronize biological clocks and, thus, reduce circadian disruption caused by shift work, jet lag, and social jet lag [135,137,138]. Palatable food, such as chocolate, scheduled for breakfast is a valuable aid in maintaining circadian synchrony and improving body weight. The present data agree with the previous literature and indicate that the thermogenic effect of a high-calorie food such as chocolate is actually different depending on the time of intake in relation to the activity–rest phase: high postprandial thermogenic response takes place during the active phase, not during the rest phase. Breakfast induces a strong postprandial thermogenesis, which induces an increase in energy expenditure [151,152,153].

A piece of chocolate a day may also modulate circadian oscillations in brain areas involved in the reward system [154,155]. It may also have effects on behavior, mood, and cognitive functions.

In a socio-historic period such as the current one, when psychophysical distress is significantly increasing, due in part to the pandemic that has led to social distancing, significant repercussions on economic stability and the ability to envision future prospects are to be expected. The intake of chocolate rich in flavanols at breakfast and during the active phase, in accordance with the rules of chrononutrition, could provide a new tool that is economical, quick, and easily applicable for all ages to support cognitive performance during periods of sleep deprivation and psychophysical stress as well as during phases of desynchronization of circadian rhythms, with a positive effect on mood, while also helping to protect our cardiovascular system and regulate our metabolism.

Future studies in selected populations will be essential to establish the correct ways of taking chocolate by defining dosages as well as timing during the day and over a lifetime.

## Figures and Tables

**Figure 1 nutrients-14-03113-f001:**
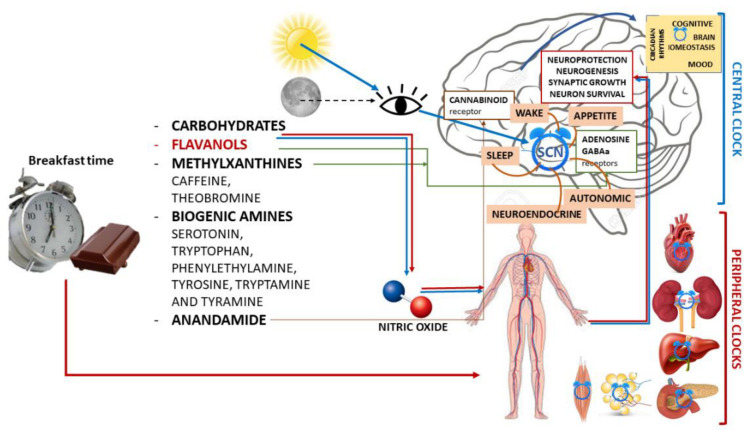
Schematic illustration of chrononutrition with chocolate for breakfast and components of chocolate with the main mechanisms of action in both the central nervous system and peripheral organs and their actions.

## Data Availability

Not applicable.

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
