# Peer review of "Cyrcadian Rhythm, Mood, and Temporal Patterns of Eating Chocolate: A Scoping Review of Physiology, Findings, and Future Directions"

_nutrients, 2022, doi:10.3390/nu14153113_

Round 1

Reviewer 1 Report

The authors summarized the role of neuromodulation, mood and sleep of flavonoid in chocolate by central clock and peripheral clocks. The manuscript is VERY interesting. These literatures were integrated to provide a chrononutrition strategy for the discovery of natural product-derived neuromodulation drugs to improve. 

Author Response

I thank the referees for their positive comments on the contribution and for appreciating the effort to integrate the available literature data and to propose an integrated approach to chrononutrition and chronotherapy with natural products.
I have corrected the duplicates in the contribution and the English language with the help of a review by native speakers through the word review mode. Please see the attachment

Reviewer 2 Report

I found this manuscript very well written and informative. It is very thorough and covers all possible physiological aspects of consuming chocolate. The only criticism I have is that the manuscript is a little bit verbose and could use a bit of a rational shortening.

The focus of the paper perfectly fits the scope of the journal. I recommend accepting it.

Author Response

(The authors gave the same response as above.)
